# The role of foreign capital and economic freedom in sustainable food production: Evidence from DLD countries

**Guoping Ding**[1☯], **Prince Asare Vitenu-Sackey**[1☯]*, **Wenshu Chen**[1☯], **Xiaofeng Shi**[2,3☯], **Jun Yan**[1☯], **Shichen Yuan**[1☯]

**1** Jiangsu University, School of Finance and Economics, Zhenjiang, Jiangsu Province, P. R. China, **2** Jiangsu University Hospital, Zhenjiang, Jiangsu Province, P. R. China, **3** Nanjing Medical University Hospital, Nanjing, Jiangsu Province, P. R. China

☯ These authors contributed equally to this work.
* pavsackey@gmail.com

**Data Availability Statement:** Data that support this study has been deposited at Mendeley data and can be found at http://dx.doi.org/10.17632/fr2fcvftpj.1.

## Abstract

In many developing countries, the deficiency in public and private investment has resulted in lower growth rates and stagnation in productivity. The need for a new paradigm of foreign investment and aid in agricultural production is becoming exigent in developing countries. Given the decline in per capita arable land, the rise in production costs, and the increase in population and urbanization, major changes in agriculture have been proposed to boost agricultural production. This present study endeavours to contribute to the existing literature by proving whether foreign capital and economic freedom could catapult food production into the much-anticipated growth. In that regard, we performed GLS with correlated disturbances, system GMM dynamic panel data estimators and D-H causality test. The study found that foreign capital as a whole plays a positive role in food production in developing and least developed countries, but FDI is insignificant in least-developed countries. Moreover, economic freedom plays a positive role in food production in least-developed countries but negative in developing countries. Policymakers and governments should create an enabling environment for sustainable food production.

## 1. Introduction

At this time and age, the paradox of hunger still exists even though the world produces more food than before—however, about 690 million of the global population sleep hungry and suffer from malnutrition [1]. In 2019, approximately 135 million of the world's population suffered acute hunger and food insecurity, precisely living in 55 countries [2]. Achieving zero-hunger in the global context is the United Nations' ultimate goal, goal number 2 of the 17 SDGs. The goal pledges to end hunger, improve nutrition, achieve food security, and promote sustainable agriculture [2]. A report from the United Nations suggests that efforts to achieve zero-hunger by 2030 have deteriorated. The report further posits that persons living in hunger could surpass the current estimate of 690 million to 800 million in 2030 if the current trend persists.

**Funding:** The author(s) received no specific funding for this work.

**Competing interests:** The authors have declared that no competing interests exist.

**Abbreviations:** SDGs, Sustainable development goals; FARA, Forum for Agricultural Research in Africa; DESA, Department of Economics and Social Affairs; DLD, Developing and Least developed countries; ILO, International Labour Organisation; FAO, Food and Agriculture Organisation.

The Coronavirus pandemic has consequently caused this havoc to persist, estimated to cause additional 130 million persons into acute food insecurity by the close of 2020 [3].

In the event of the pandemic's impact on food supply and the agricultural sector as a whole, pragmatic and mitigating measures ought to be instituted to curtail the shocks witnessed to keep food supply systems active and alive in order to relieve the vulnerable and the poor from the devastating effects [4]. The problem of acute food supply and insecurity can be found in the developing and least-developed countries due to the economic downturn in those regions, specifically, Asia and Sub-Saharan Africa [2]. Interestingly, a larger number of the population in those regions is into agriculture for their livelihoods. Due to financial constraints, these persons cannot inject funds to turn around their fortunes to alleviate poverty [5]. According to FAO [6] and Chichava et al. [7], food production in developing countries is inadequate and requires a significant agricultural investment [8] due to agricultural technology constraints and severe weather conditions. For this purpose, development agencies and organizations have been urgently resorted to addressing these problems with foreign financial assistance [9, 10]. According to the World Food Programme [2], massive governments support, developmental assistance (humanitarian aid), and investment from the private sector could significantly alleviate the hardship in the agricultural food production sector to create innovative and lasting solutions [5, 8, 11]. Furthermore, the Food and Agriculture Organisation [4] suggests that governments should bridge the gap between the rich and the poor by providing social safety net programs to alleviate the vulnerable and poor persons' plights.

Countries relying on imported foodstuffs are particularly susceptible to slowing trade flows, particularly if their currencies are weakening as they have been. Although consumer food prices are prospective to increase all over, their effect is more negative where food prices are abrupt, serious, and unpredictable, where food expenses constitute a more significant share of household expenditures, and where surges may have a longer-term impact on human development and potential economic growth. Governments should reassess their trade and tax policy choices and their possible effects immediately and work in concerted efforts to create an enabling climate for food trade [4, 11, 12]. Attending to these issues garnered towards economic freedom, and economic freedom translates into poverty alleviation. Moreover, poverty alleviation is proportionate to hunger alleviation [3].

An average annual investment of $209 billion is needed, according to FAO forecasts, to achieve the expected demand for agricultural food production of 93 developing countries by 2050 [13]. Because most less developed countries have limited agricultural investment potential, international agricultural investment ought to be used to meet their developmental needs [14]. In the rural and other sectors of the economy, good upstream and downstream linkages also stimulate growth and revenue generation. None of these nations could then make significant progress in poverty reduction, economic expansion, and higher food security unless they support the agricultural sector's human capital and future productive ability to increase its involvement in social and economic development to be specific [15, 16]. Today, in low-income countries, agriculture accounts for just 23% of GDP, but in middle-income and high-income countries, it accounts for 10% and 2%, respectively [17]. Only in low-income countries does the share of jobs in the agricultural sector surpass 60 percent [18]. Accordingly, DESA [19] and FARA [20] accepted that the contribution of agriculture to GDP had decreased significantly over time in numerous states of the world due to underinvestment in this particular economic sector. The deficiency of public and private investment in many developing countries has headed to lower output growth rates and stagnation in productivity [21]. In most developing countries, the need for a different paradigm of foreign investment and aid in food production is becoming exigent. Given the decline in per capita arable land, the rise in production

costs, and the increase in population and urbanization, significant changes in agriculture have been proposed to boost agricultural production [22].

Many scholars see foreign direct investment as an agricultural production engine because it is accompanied by technological transfers, managerial and technological skills, and know-how to support farmers [8, 11, 22–25]. In addition to the role of foreign direct investment, a different aspect of the studies emphasize the importance of foreign aid in this sector's growth. While developing countries are agricultural economies with long-standing foreign capital channeled into them, there is a limited plethora of experimental evidence dedicated to foreign capital's capacity to promote the sector. Opposed to what the current paper advocates, there is a vast literature that argues that transnational capital nestled in a global food regime promotes capital accumulation in the country side on one hand, and generates distress among petty commodity producers on the other hand. In the face of the distress at the countryside, sustainable food production may not be possible. Further, this alternative strand of literature argues that the transnational capital amplifies the uneven development that is taking place at a global scale [26–28]. Another apprehension associated with inflows of foreign capital into the developing country agriculture relates to the alteration of cropping patterns in these countries. Foreign capital may promote a cropping pattern that aligns with the need/demand of the global food market. With a number of countries adopting a similar cropping pattern, the terms of trade (due to excess supply of similar crops in the global market) may go against the developing country farmers [29, 30]. Given the above assertions, the study intends to find an answer to this seeming question that "*what role does foreign capital play in food production in developing and least-developed countries considering the intervening role of economic freedom?*" The novelty of this study stems from:

1. To the best of our knowledge, this is the first attempt to link foreign capital and economic freedom to food production. This study aims to find answers to the unanswered question by adopting several econometric techniques to understand this phenomenon critically.

2. Agriculture is considered an essential element in any economy's growth, reducing hunger, providing sustenance, and alleviating malnutrition, perhaps critical in developed and developing countries [31]. Correspondingly, Djokoto [32] has shown that agriculture is considered an essential source of jobs in developing countries. Developing countries constitute the top beneficiaries of foreign aid, with US$765 billion in official developmental assistance inflows in 2015 [33]. Therefore, the focus of this study is on developing and least-developed countries. It is primarily due to the prominence of agricultural productivity in these economies' clusters, hence emphasizing developing and least-developed countries. It is evident that the largest share of the populace living in developing countries is related to agricultural production directly or indirectly within their economies [34].

3. This present study endeavours to add to the existing scholarly works by illustrating whether foreign capital and economic freedom could catapult food production into the much-anticipated growth. However, we use second-generation econometric approaches such as Pesaran [35, 36] cross-sectional dependence test; Pesaran [37] and Westerlund [38] cointegration tests, contemporaneous correlation estimator (generalized least square with correlated disturbances), Arellano and Bover [38] and Blundell and Bond [39] system GMM dynamic panel data estimation method to identify the long-run coefficients, and Granger causality to ascertain the direction of a causal relationship.

This study consists of six sections, namely; (1) introduction, (2) literature review, (3) data and methodology, (4) empirical analysis and results' presentation, (5) findings discussion, and (6) conclusion and policy implication.

## 2. Literature review

### 2.1 Foreign direct investment and sustainable food production

According to the neoclassical theorists, foreign direct investment inflows channeled from developed countries into developing countries have been established as a crucial element in fostering economic growth; when states do not interfere in the markets, apparently economic development is achieved [40–43]. Host countries have been pinpointed to enjoy the advantage of foreign direct investment inflows, such as raising the technological level of a nation, new opportunities for jobs, tax revenues, providing a new source of capital, and access to international markets [44, 45]. In the current environment, foreign investment forms an integral part of economic growth and economic development as many countries step towards globalization. Over the past few years, almost all developing countries have seen an increase in foreign investment interest in the agricultural sector [46].

Foreign Direct Investment (FDI) inflows to agriculture, forestry, and fisheries (AFF) rose from $1.2 billion to $1.7 billion from 1997 to 2011, with considerable fluctuations on a year-to-year basis. In 2007, FDI inflows to AFF peaked when they hit a record $7.3 billion in 2007 (all prices are in constant 2005 U.S. dollars). Over the same period, overall FDI inflows doubled, driven primarily by cross-border mergers and acquisitions (M&A), from $560 billion in 1997 to $1.1 trillion in 2011, with a high of $1.6 trillion in 2007 [47]. As a result of the economic downturn and a rapid fall in stock prices in early 2001, the value of mergers and acquisitions decreased by 48 percent in 2000–2001 and 38 percent in 2001–2002. In 2004 (21 percent annual increase), 2005 (35 percent), 2006 (48 percent), and 2007 (35 percent), FDI inflows increased again but fell in 2008 and 2009 due to credit restriction and an end to high stock prices. With improving economic and financial conditions, the value of mergers and acquisitions increased by 36 percent in 2010. Developing regions accounted for almost 90% of all inflows between 2004 and 2011, or a total of $23.6 billion, as the key target for Agriculture Forestry and Fisheries FDI inflows to developing countries. The biggest beneficiaries were countries in Asia and the Pacific (A&P), earning 45% of the overall inflows to AFF. However, during this time, inflows to Africa proliferated, increasing from 4 percent to 25 percent of total FDI inflows to the Agriculture Forestry and Fisheries sector [48].

Empirically, a study analyzed the effect of sectoral foreign aid on output across economic sectors in Latin America. Positive contributions of FDI to agriculture, services, and manufacturing sectors performance linked to the productivity of agricultural, forestry, and fisheries sector were established [49]. Moreover, another addresses the impact of agriculture growth in describing the association between FDI and food security, using unbalanced panel data from 1995 to 2009 of 55 developing countries. They found that through the intervening effect of agricultural production, FDI increases food security [50]. Most recently, in selected Islamic Cooperation Organization (ICO) countries from 2003 to 2012 (Specifically, Oman, Malaysia, and Brunei), the study explored the effects of foreign direct investment determinants (poverty, infrastructure, inflation, market size, and exchange rate) in the agricultural production industry. Their conclusion contended that foreign direct investment directly links agricultural food production [51]. Dhahri & Omri [52, 53] observed in their studies that foreign direct investment plays a positive and significant role in sustainable food production in developing countries. On the backdrop of 50 developing countries, they utilized the fixed effect model estimator for both studies. Moreover, they contend that the linkage between aid and foreign direct investment is inherent and doubles food production growth.

## 2.2 Foreign aid and sustainable food production

Numerous researches share a similar inherent hypothesis that foreign aid has boosted agricultural development in developing countries [54]. The overall portion of foreign aid apportioned to the agricultural production sector has declined suddenly by almost 1/3 in real terms from 1995 to 2015, despite substantial evidence demonstrating the vibrant function of aid focused on the agricultural sector [52]. The majority of donor interest and donor capital have been geared towards investment in structural change over the past 21 years. In the early 1995s, agricultural aid constituted 9% of total official developmental aids, falling to 4% at the close of 2002, when aid reached the utmost level and reaching nearly 6% of all ODA in 2015. The share of sectoral aid highlights this decline in agricultural aid share in developing countries [52].

According to other scholars, this drop in agricultural aid share can have adverse side-effects by constraining agriculture growth and development, especially in low-income countries where improvements are needed [55–57]. Recently, there has been a resurgence of interest among organizations, donors, scholars, and policymakers in the agricultural sector concerning foreign aid. In general, a controversial issue has been the connection between foreign aid and agricultural production. Specifically, no agreement has emerged yet, and the empirical findings are most unusual and diverse. In light of this, Kaya et al. [57] investigated the impact on per capita income of official developmental assistance channeled into the agricultural industry using the generalized moment method of moment (GMM) for 66 developing countries. Their study contended that when official developmental assistance is geared towards the agricultural industry, it could positively and significantly impact economic growth in the short run in developing countries.

Furthermore, Kaya et al. [58] used annual data from 1974 to 2005 to evaluate the effect of agriculture aid on agricultural production in developing countries. They observed a positive correlation between agricultural productivity growth and agricultural aid, noting that foreign aid dedicated to the development goal would achieve its targets if aid is aimed at developing countries' agricultural sectors. Alabi [59] examined the connection between agricultural aid and agricultural production growth; they observed that aid positively contributes to agricultural productivity.

Interestingly, the study revealed that bilateral aid has a more potent linkage to agricultural production than multilateral aid regarding agriculture growth. According to Dhahri and Omri [30], total foreign aid (Non-investment aid, Social & infrastructural aid, Agriculture aid, and investment aid) is proportionally related to agricultural production. Perhaps its connection between foreign aid and sustainable food production is very positive and significant. Their study focused on 50 developing countries over the period 1995 to 2015. Moreover, Dhari & Omri [31] buttress their observation in a previous study that aid tends to increase agricultural production. However, some sectorial aids insignificantly contribute to sustainable agricultural production, thus non-investment aids and investment aids when foreign direct investment intervenes.

Contrary to the above contentions, some past prolific economists argued that foreign assistance (aid) might negatively impact incentive mechanisms and increase dependence in developing countries [60–62]. Moreover, Rodrik and Subramanian [63], Rajan and Subramanian [64], Kanbur et al. [65], and Younger [66] also argued that foreign aid would negatively impact the productivity of a country and the management ability of the government. In furtherance, Friedman [67] and Bauer [68] suggested that foreign assistance (aid) could be inefficiently used to support the political elite and increase corruption in the government. Upon weighing these scholars' arguments, evidence suggests no consensus about the exact connection between foreign official development assistance and sustainable food production.

## 2.3 Economic freedom and sustainable food production

What causes economies to develop is one of the most enduring problems in economics. Adam Smith's book published in 1776 titled "An Inquiry into the Existence and Causes of the Wealth of Nations," explicitly indicates that Smith's primary concern was it causes prosperity. He suggested that free markets, private property rights protection, and a limited government role contribute to growth in the economy. Economic freedom vehemently contributes to the growth of an economy. He further supported low taxes and government spending, low tariffs, and protected private property rights to facilitate foreign trade to enable resource distribution in open markets. In other words, Smith argued that the economy would expand and flourish if a business climate was established and sustained [69].

The view of Adam Smith pays minimal attention to inputs by concentrating on climate conduciveness to economic growth. However, Smith also admitted that the market's unseen hand would do an efficient job of allocating capital if it were enabled to function under an atmosphere of economic freedom. If this favorable atmosphere is established, public policy should not concern itself with creating resources, introducing technology, or developing a trained labor force. The economy would encourage investment and provide incentives for both employees to develop marketable skills and innovative technologies. The right environment will attract the right inputs, but it will not build the right environment to provide the right inputs. Growth will follow if the growth strategy focuses on building a climate of economic independence. Periodically, when the right setting is not considered, growth will not happen [69, 70].

There is an indication that even in countries where political freedom is restricted, economic freedom contributes to economic growth. The opposite is not the case: without economic freedom, political freedom does not bring prosperity. Consequently, it is crucial that developing democracies facilitate free markets, provide a stable currency, safeguard property rights, and lessen government role in the economy. It is also proof that higher-income countries tend to have political freedoms, more democratic, and uphold civil rights. Indirectly, thus, economic freedom contributes to political liberty [69, 71–73]. Also, evidence strongly suggests that growth will not take place without an atmosphere of economic freedom. For an economy to expand, economic freedom requires several aspects, all of which must be in place. There is a need for an economy to have secure private property rights, a stable monetary system, low taxes, minimal government interference, an impartial legal system, and low international exchange barriers. If all of these things are absent, there will be no development in the economy [69, 70].

The food production sector is fraught with difficulties. In solving these problems, the recent announcements are a significant beginning. Focusing on the three major decisions of private individuals—what to plant, how much to invest, and what to store—and thinking about the opportunity climate for better decision-making is essential. Far less political participation, improved warehousing, strengthened derivatives markets, foreign trade, property rights on agricultural land, taxation of food production and national trade are involved in the path forward [71]. The futures market is the most significant input into this. The futures market allows private individuals to research the food production sector and make potential price predictions. Private individuals would benefit from a well-functioning futures market by comparing futures prices, and sowing and storage will move to commodities where potential shortages are evident. This environment needs to eliminate tight regulations on futures trading, integration with global futures markets, and the usual agenda of having the financial markets work well [71].

In other studies, Aghion et al. [72] investigated the relationship between economic volatility and growth by connecting productivity-enhancing investment and credit constraints. In their study, they considered the role of economic institutions where economic freedom was proxy by property right protection from Caselli and Wilson [73] compiled data. On the other hand, several studies used institutional quality to represent economic freedom as a measure of stronger institutions [74, 75]. In a recent study, Kouton [71] utilised the economic freedom index data compiled by the Heritage foundation to analyse the association between economic freedom and inclusive growth in the sub-Saharan Africa. The study contends that stronger institutions substantially influence inclusive growth where it suggested that the economic freedom index represents the role of socio-economic institutions keenly. Meanwhile, economic freedom serves as a crucial factor which elucidates growth and stagnation through two paths. Firstly, with emergence of new designs and development of advanced technologies through technological advancement hence it becomes an essential determinants of growth. Secondly, the enforcement of laws through property right protection and corruption control as well as openness of the economy and investment freedom serve as a crucial growth determinants [76].

To the best of our knowledge, no study has been conducted considering economic freedom with foreign capital and food production. Therefore, it is pertinent to ascertain whether economic freedom significantly impacts food production in developing and least-developed countries.

## 3. Theoretical model and econometric technique

### 3.1 Theoretical model

In 1928, the economists Cobb and Douglas proposed an economic theory dubbed "Cobb-Douglas production function" [77]. This theory propounds a relationship between the output and inputs of firms or countries. The production function can be found as:

$$Y = AK^{\alpha}L^{\beta} \tag{1}$$

In the production function Eq (1), Y represents total output of production, A represents total factor productivity (technological advancement and efficiency), K represents total capital inflow input, L represents total labor participation, $\alpha$ and $\beta$ represent the elasticities of labor and capital linked to output, simultaneously.

In retrospect, scores of studies evaluated the Cobb-Douglas function by considering output as a gross domestic product (real GDP) to elucidate its association with other inputs or factors [78, 79]. Despite the importance of agriculture to the growth of an economy through production, perhaps it directly links as inputs [80–82]. In light of this, the exclusion of agriculture in the output production function is not conceptual. The production function of output is determined by capital stock, agriculture production, and labour force [83, 84]. Total factor productivity in a conceptual context refers to the intense and efficient use of inputs in the production process. It is mostly related to multifactor productivity and can be referred to as the level of knowledge or technology under certain assumptions. Total factor productivity is sometimes referred to as other factors such as monetary shocks, government integrity or effectiveness, judiciary effectiveness, etc. but not necessarily as a measure of technological input. In essence, the presence of crime, computer viruses, corruption, political instability, judiciary ineffectiveness negatively impacts capital stock (K) and labour (L) in the measure of total factor productivity or efficiency of an economy [85].

In the study's concept, economic freedom represents total factor productivity due to its measure of an economy's efficiency. The index of economic freedom encompasses nine indicators in measuring economic freedom thus property rights, financial freedom, government

spending (government size), investment freedom, monetary freedom, fiscal freedom (fiscal health), freedom from corruption (judicial effectiveness), Labour freedom, and trade freedom [70]. Nonetheless, countries characterized by higher economic freedom levels grow exponentially. They usually have greater longevity and attain higher per capita incomes than countries with low economic freedom [86]. Capital stock emanating from foreign direct investment and foreign aid tends to thrive in countries with high economic freedom. Based on this assumption, the study proposes the model below on the Cobb-Douglas production function's modified form.

$$Y = f(FDI, AID, EFIO) => Y = EFIO^{\alpha} FDI^{\beta} AID^{\emptyset} \qquad (2)$$

Where Y represents food production, FDI represents foreign direct investment, AID represents foreign aid, EFIO represents economic freedom, and $\alpha$, $\beta$, $\emptyset$ are the output elasticities of economic freedom, foreign direct investment, and foreign aid. In furtherance, other variables are included in the model as control variables to support the analysis of the relationship between food production and foreign capital (foreign direct investment and foreign aid). Hence, population growth, economic growth, and consumer price index are selected as control variables in support of literature from Gardi et al. [87] and Galinato and Galinato [88].

After taking natural logarithm of the variables, the econometric model for the study is constructed as:

$$lnHA_{it} = \beta_0$$

$$+ \beta_1 foreign\ capital_{it} \begin{bmatrix} lnFDI \\ lnAID \end{bmatrix} + \beta_2 lnEFIO_{it} + \beta_3 lnY_{it} + \beta_4 lnPOPG_{it} + \beta_5 lnCPI_{it} + \varepsilon_{it} \quad (3)$$

The equation above simplifies that food production is the function of foreign capital inflows, economic freedom, economic growth, population growth, and consumer price index. However, in Eq (3), lnHA represents food production, lnFDI, and lnAID are proxies for foreign capital represented by foreign direct investment and foreign aid, respectively. Also, lnEFIO represents the economic freedom index, lnY represents economic growth, lnPOPG represents population growth, and lnCPI represents the consumer price index. On the other hand, $\beta_1$ to $\beta_5$ are coefficients of the parameters (foreign capital, economic freedom, economic growth, population growth, and consumer price index), where $\varepsilon$ represents the error term, $\beta_0$ represent the constant term or intercept of the model, i represents the cross-section of 71 countries sampled for the study, and t represents the period from 1995 to 2018 as the sample period.

## 3.2 Econometric technique

The sections below outline the econometric approaches followed:

**3.2.1 Unit root and cross-sectional dependence test.** According to Asteriou [89], there is an assumption that long-run parameters are likely to experience cointegration among a set of I (1) variables. Perhaps, macroeconomic variables included in the model could have unit roots; hence they would not be stationary [90]. However, it becomes essential to perform a unit root test to check for the stationarity status of the variables selected for statistical reliability. That notwithstanding, four-unit root tests were considered in this study, thus Pesaran CIPS and CADF unit root tests [91]. Subsequently, the study performed a cross-sectional dependence test to unravel the contemporaneous correlation among the selected variables. In so doing, Pesaran [36] test was performed.

**3.2.2 Cointegration test.** The confirmation of no unit root but cross-sectional dependence of the selected variables pave the way for further test. However, at this step, Pedroni [37] and

Westerlund [38] second-generation panel cointegration are performed to evaluate the long-run relationship between the dependent and the independent variables. Cointegration assumes that the alternative hypothesis I(1) suggests cointegration evidence between the dependent and independent variables. On the contrary, the null hypothesis I(0) suggests that the variables are not cointegrated. Therefore, $H_1$: $\beta < 0$ is expected to be accepted and Ho: $\beta_1 = 0$ to be rejected at 5% or 1% significance levels.

**3.2.3 Correlation matrix.** Invariably, the correlation matrix produces the correlation coefficients of the variables and displays the correlations' sign. Moreover, the coefficients displayed reveal whether there is multicollinearity or not. According to Sun et al. [92], evidence of coefficients of -/+0.70 of two or more independent variables with the dependent variables suggest a multicollinearity presence in the model. Therefore, the correlation matrix computed would reveal the model's status, whether there is a presence of multicollinearity or not.

**3.2.4 Long-run estimation.** In this step, two regression analyses were performed using panel generalized least square with correlation disturbances (GLS) and system dynamic generalized method of moment dynamic panel data estimator [34, 35]. The system dynamic GMM two-step approach we tend to provide solutions to common issues such as simultaneity, endogeneity, and heterogeneity (bidirectional causality problems among the variables) which meets the need for panel data analysis. Using the GLS, we intend to estimate the unknown parameters in the panel that would be correlated. Since GLS is the best estimator to resolve the problem of autocorrelation disturbances and serial correlation, Koreisha and Fang [93] observed that the estimator could correctly identify parameters that are inefficiently estimated in the procedure. However, the system dynamic panel data estimator allows the first differenced errors of the endogenous (dependent) variable to use moment conditions. In contrast, the level equation utilizes the dependent variable as an instrument. Moreover, the independent variables (exogenous) are considered as instruments in the first-differenced equation. The method resolves simultaneity, endogeneity, and cross-sectional heterogeneity by applying the country-specific effects to reduce the unobserved variability.

**3.2.5 Pairwise Dumitrescu Hurlin panel causality test.** With reference to Shabbaz et al. [94], the causality test reveals more information regarding the causal relationship's direction among selected variables for policy implications. Dumitrescu and Hurlin [95] contend that the first difference stationarity variables require a causality test to check the causal linkage direction. Therefore, the study performed a causality test to ascertain whether the variables are unidirectional or bidirectional related.

## 3.3 Data

The study's data were sourced from the World Bank's World Development Indicators and HeritageFoundation.org from 1995 to 2018. The data covers 71 countries comprising of 40 least-developed countries and 31 developing countries (see S1 Appendix for list of countries). The choice of the 71 countries and study period are due to data availability. The study's dependent variable is food production index as its proxy, and foreign capital with foreign aid and foreign direct investment as proxies. Also, economic freedom with economic freedom index as a proxy. Moreover, economic growth, population growth, and consumer price index are used as control variables. More details about the variables can be found in Table 1.

# 4. Empirical analysis and results

## 4.1 Descriptive statistics

Table 2 presents the descriptive statistics of the selected variables for the study. In order of distribution, it can be reported that the variables are not normally distributed. Evidence

**Table 1. Variables description and sources (Source: Authors' construct).**

| Indicator | Variable | Description | Source |
|---|---|---|---|
| lnHA | Food production | Food production index (2004–2006 = 100) Food production index covers food crops that are considered edible and contain nutrients. Coffee and tea are excluded because, although edible, they have no nutritive value. | World Bank—World Development Indicators |
| | Foreign Capital: | | |
| lnFDI | Foreign aid | Net official development assistance and official aid received (constant 2015 US$) | World Bank—World Development Indicators |
| lnAID | Foreign direct investment | Foreign direct investment, net inflows (BoP, current US$) | World Bank—World Development Indicators |
| lnY | Economic growth | GDP per capita, PPP (constant 2011 international $) | World Bank—World Development Indicators |
| lnPOPG | Population growth | Population growth (annual %) | World Bank—World Development Indicators |
| lnCPI | Consumer Price Index -Inflation | Consumer price index (2010 = 100) | World Bank—World Development Indicators |
| EFIO | Economic freedom | Economic freedom index—Property Rights, Judicial Effectiveness, Government Integrity, Tax Burden, Government Spending, Fiscal Health, Business Freedom, Labor Freedom, Monetary Freedom, Trade Freedom, Investment Freedom, Financial Freedom | Heritagefoundation.org |

from the Jarque-Bera test of each variable confirms a p-value less than 5% hence not normally distributed. This implies that the ordinary least square regression method would not be appropriate for estimating the long-run coefficients. Moreover, the variables are positive in Kurtosis but negatively skewed. Table 2 shows that the average growth rate of food production in least-developed and developing countries is 4.645% annually. In comparison, economic growth (lnY) grew at an annual average of 8.151% during the sample period. With respect to foreign capital thus foreign direct investment inflows and foreign aid grew at 18.150% and 19.536% annually, respectively. Moreover, the sample period's consumer price index grew at an annual index score of 4.164, and the population growth rate stood at 0.527%. In relation to economic freedom, the average index score increased by 3.800 annually, signifying improvement in the economic freedom of people living in least-developed and developing countries.

**Table 2. Descriptive statistics.**

| | lnHA | lnFDI | lnAID | lnY | lnCPI | lnPOPG | EFIO |
|---|---|---|---|---|---|---|---|
| Mean | 4.645 | 18.150 | 19.536 | 8.151 | 4.164 | 0.527 | 3.800 |
| Median | 4.626 | 19.318 | 19.808 | 8.130 | 4.427 | 0.700 | 4.009 |
| Maximum | 7.609 | 24.518 | 23.151 | 9.956 | 7.916 | 2.094 | 4.349 |
| Minimum | 3.792 | 0.000 | 0.000 | 6.301 | -7.265 | -4.564 | 0.000 |
| Std. Dev. | 0.271 | 4.843 | 1.975 | 0.858 | 1.145 | 0.669 | 0.871 |
| Skewness | 1.548 | -2.818 | -6.255 | -0.009 | -3.312 | -2.056 | -3.981 |
| Kurtosis | 19.138 | 10.987 | 62.221 | 2.139 | 18.590 | 10.533 | 17.502 |
| Jarque-Bera | 19170.180 | 6785.375 | 260117.800 | 52.601 | 20371.630 | 5229.607 | 19433.990 |
| Probability | 0.000 | 0.000 | 0.000 | 0.000 | 0.000 | 0.000 | 0.000 |
| Sum | 7915.036 | 30927.28 | 33289.85 | 13888.61 | 7095.803 | 898.1512 | 6475.805 |
| Sum Sq. Dev. | 125.2474 | 39950.71 | 6643.603 | 1254.306 | 2230.758 | 762.9525 | 1290.99 |
| Observations | 1704 | 1704 | 1704 | 1704 | 1704 | 1704 | 1704 |

Source: Authors' construct

## 4.2 Panel unit root tests

Unit root tests fish out the stationarity status of the selected variable to avoid spurious regression coefficients estimation. The unit root test assumption suggests that there is evidence of unit root in the variables; hence, they are not stationary. Among a set of I(1) variables, the long-run parameters are most likely prone to cointegrated relationships; hence the expectation is that the macroeconomic variables selected in the study could exhibit unit root in the process. This warrants the need to test for unit root, and test results can be found in Table 3. At level form, most of the variables could not reveal stationarity. Nevertheless, performing the tests in the first difference, all the variables confirmed stationarity at a 1% significance level.

Also, in Table 3, the tests for cross-sectional dependence (CD) of the variables are reported. Evidence from the results confirms that all the variables have cross-sectional dependence except POPG, which could not substantiate evidence of cross-sectional dependence in the least-developed countries sub-sample. Specifically, at a 1% significance level, the assumption that all the variables are cross-sectional dependence is substantiated.

**Table 3. Panel unit root test (Source: Authors' construct).**

|  | LEVEL | | FIRST DIFFERENCE | | |
|---|---|---|---|---|---|
| **All Sample** | **CIPS** | **CADF** | **CIPS** | **CADF** | **CD-TEST** |
| LNHA | -3.362*** | -6.909*** | -5.337*** | -16.877*** | 144.234*** |
| LNFDI | -3.445*** | -10.214*** | -5.360*** | -21.148*** | 69.903*** |
| LNAID | -2.435*** | -0.774 | -5.279*** | -17.529*** | 37.793*** |
| LNPOPG | -2.767*** | -3.306*** | -3.304*** | -10.494*** | 19.452*** |
| LNCPI | -2.098 | -12.044*** | -3.710*** | -14.146*** | 224.447*** |
| LNY | -1.844 | 0.408 | -4.118*** | -9.783*** | 145.92*** |
| EFIO | -3.361*** | -7.674*** | -5.160*** | -16.417*** | 38.134*** |
| **Developing Countries** | CIPS | CADF | CIPS | CADF | |
| LNHA | -3.101*** | -0.800 | -5.359*** | -9.427*** | 45.063*** |
| LNFDI | -3.640*** | -7.161*** | -5.590*** | -15.351*** | 27.857*** |
| LNAID | -2.680*** | -0.797 | -5.163*** | -11.571*** | 3.713*** |
| LNPOPG | -2.956*** | -5.792*** | -2.827*** | -7.450*** | 24.148*** |
| LNCPI | -1.626 | 0.613 | -3.558*** | -4.280*** | 100.182*** |
| LNY | -1.809 | 0.525 | -3.993*** | -5.655*** | 78.127*** |
| EFIO | -2.542*** | -3.175*** | -4.667*** | -9.675*** | 8.377*** |
| **Least-Developed Countries** | CIPS | CADF | CIPS | CADF | |
| LNHA | -3.479*** | -5.260*** | -5.526***a1 | -12.233*** | 99.683*** |
| LNFDI | -2.905*** | -4.801*** | -5.086*** | -13.926*** | 41.984*** |
| LNAID | -3.005*** | -4.397*** | -5.430*** | -14.401*** | 40.733*** |
| LNPOPG | -2.165** | -9.854*** | -2.512*** | -13.069*** | 0.317 |
| LNCPI | -2.258*** | -3.884*** | -3.934*** | -9.032*** | 122.519*** |
| LNY | -1.272 | 0.562 | -3.733*** | -7.940*** | 67.55*** |
| EFIO | -3.952*** | -6.414*** | -5.378*** | -11.489*** | 34.315*** |

Note

*** indicates 1% significance level

** indicates 5% significance level. LLC = Levin, Lin & Chu test, IPS = Im, Pesaran & Shin test, ADF and PP test = Maddala & Wu tests. CD = Cross-sectional dependence. lnHA = Hunger alleviation, lnFDI = foreign direct investment, lnAID = foreign aid, lnY = gross domestic product per capita, lnCPI = consumer price index, lnPOPG = population growth, EFIO = economic freedom index

**Table 4. Panel cointegration test (Source: Authors' construct).**

| Westerlund | All sample | | Developing Countries | | Least-developed countries | |
|---|---|---|---|---|---|---|
| | Z-value | Robust P-value | Z-value | Robust P-value | Robust P-value | Robust P-value |
| Gt | -15.786*** | 0.000 | -14.279** | 0.020 | -8.995** | 0.010 |
| Ga | -5.836*** | 0.000 | 6.754** | 0.010 | 9.268** | 0.010 |
| Pt | -17.448*** | 0.000 | -13.997** | 0.030 | -20.586*** | 0.000 |
| Pa | -8.954*** | 0.000 | 5.212** | 0.040 | 2.873*** | 0.000 |
| **Pedroni** | All sample | Developing Countries | Least-developed countries | | | |
| Panel v-Statistic | 1.626** | 1.890** | 0.111 | | | |
| Panel rho-Statistic | 2.312 | 2.729 | 0.103 | | | |
| Panel PP-Statistic | -13.900*** | -6.799*** | -13.472*** | | | |
| Panel ADF-Statistic | -15.517*** | -8.536*** | -13.831*** | | | |
| Group rho-Statistic | 4.741 | 2.665 | 3.970 | | | |
| Group PP-Statistic | -19.214*** | -13.157*** | -14.015*** | | | |
| Group ADF-Statistic | -15.177*** | -11.427*** | -10.160*** | | | |

Note

*** indicates 1% significance level

** indicates 5% significance level

## 4.3 Cointegration test

Estimating for cointegration relationship among variables reveals their long-run relationship either they are cointegrated or not. In the study's quest to check for the long-run relationship between the dependent and the independent variables, the Pedroni [37] and Westerlund [38] second-generation panel cointegration test were performed (see Table 4). The results show that all the variables are cointegrated at a 1% and 5% significance level in all the Pedroni and Westerlund test samples. Therefore, the null hypothesis suggests no cointegration evidence; hence, the assumption that there is no long-run relationship among the variables is rejected.

## 4.4 Correlation matrix

The correlation matrix provides two-fold insightful information. Firstly, correlation coefficients and signs. Secondly, information on collinearity or multicollinearity of the independent variables with the dependent variable. Table 5 displays the outcome of the correlation matrix.

**Table 5. Correlation matrix (Source: Authors' construct).**

| Correlation | | | | | | | |
|---|---|---|---|---|---|---|---|
| Probability | lnHA | lnFDI | lnAID | lnY | lnCPI | lnPOPG | EFIO |
| lnHA | 1 | | | | | | |
| lnFDI | 0.128*** | 1 | | | | | |
| lnAID | 0.025 | 0.100*** | 1 | | | | |
| lnY | 0.145*** | 0.113*** | -0.353*** | 1 | | | |
| lnCPI | 0.431*** | 0.094*** | 0.035 | 0.124*** | 1 | | |
| lnPOPG | -0.021 | -0.046* | 0.264*** | -0.444*** | -0.045* | 1 | |
| EFIO | 0.067** | 0.134*** | 0.110*** | 0.121*** | 0.125*** | -0.043* | 1 |

Note

*** indicates 1% significance level

** indicates 5% significance level

* indicates 10% significance level. lnHA = Hunger alleviation, lnFDI = foreign direct investment, lnAID = foreign aid, lnY = gross domestic product per capita, lnCPI = consumer price index, lnPOPG = population growth, EFIO = economic freedom index

In an account of the correlation between the dependent and the independent variables, foreign direct investment inflows (lnFDI), economic growth (lnY), consumer price index (lnCPI), and economic freedom (EFIO) are positively and significantly correlated with hunger alleviation (lnHA) at a 1% and 5% significance levels, respectively. However, foreign aid (lnAID) and population growth (lnPOPG) are insignificantly correlated to food production (lnHA).

The study reports no problem of collinearity or multicollinearity among the independent variables with the dependent variable. Evidence from Table 5 suggests that the variable with the highest correlation coefficient is consumer price index (lnCPI); thus, 0.431, followed by economic growth (lnY) 0.145. However, these correlation coefficients fall below the correlation coefficient threshold of -/+0.70 to be considered high correlations. In essence, the study rejects the assumption that there is evidence of multicollinearity among the independent variables, which could cause heteroskedasticity in the regression estimation.

## 4.5 Generalised least square with correlated disturbances and system GMM dynamic panel data estimations

To estimate the long-run effects of foreign capital (foreign aid and foreign direct investment) on food production, we utilized the generalized least square with correlated disturbances and system generalized method of moment dynamic panel data estimation methods to estimate the level of heterogeneity across the panels, and show the relationship between the dependent and the independent variables. At this stage, the study categorized its analyses into sub-samples of all sampled countries, developing and least-developed countries. Table 6 presents the results of the analysis for all samples. The results of the generalized least square with correlated disturbances suggest that foreign direct investment positively and significantly impact food production in all sample group and developing countries sample. Specifically, a percentage point increase in foreign direct investment could significantly increase food production by 0.003% and 0.005% at a 1% significance level, respectively. Meanwhile, the least-developed countries' results depicted an insignificant coefficient, which implies that increasing foreign direct investment in the least-developed countries could not significantly increase food production. On the other hand, foreign aid showed consistent, significant, and substantial coefficient magnitudes in all sampled groups. To be specific, a percentage point increase in foreign aid seemingly lead to an increase in food production by 0.006%, 0.001%, and 0.045% at a 1% and 10% significance level in all sampled groups, respectively. These findings substantiate the prominence of foreign capital in developing and least-developed countries. More importantly, it further highlights the urgency to inject more foreign capital into the food production sector in less-privileged economies with less access to capital to expand their food production efforts.

The findings depict an inconsistent outcome from its intervening efforts in food production to account for economic freedom. In particular, economic freedom negatively and significantly intervenes in the relationship between foreign capital and food production (hunger alleviation) in developing countries and being substantiated in our all sampled group. Contrary to this finding, economic freedom plays a positive role in food production in least-developed countries. Our findings suggest that a percentage point improvement in economic freedom could significantly increase food production by 0.025% at a 5% significance level. Nevertheless, for all sampled countries and developing countries' sample, a percentage point improvement in economic freedom could inversely impact food production (hunger alleviation). Notably, our findings suggest that economic growth and consumer price index are essential in improving food production sustainably. Specifically, a percentage point increase in economic growth could increase food production by 0.036%, 0.106%, and 0.036% across the samples at a 1% significance level, respectively.

**Table 6. Long-run estimations (Source: Authors' own estimations).**

|  | GLS |  |  | System GMM DPD |  |  |
|---|---|---|---|---|---|---|
|  | All sample | Developing | Least developed | All sample | Developing | Least developed |
| L. lnHA |  |  |  | 0.185 | 0.103 | 0.521 |
|  |  |  |  | (3.45)*** | (3.10)** | (15.64)*** |
| lnFDI | 0.003 | 0.005 | -0.002 | 0.002 | 0.006 | -0.002 |
|  | (10.42)*** | (42.99)*** | (-0.87) | (3.53)*** | (13.31)*** | (-0.83) |
| LNAID | 0.006 | 0.001 | 0.045 | 0.002 | 0.002 | 0.031 |
|  | (4.91)*** | (1.90)* | (5.23)*** | (1.80)* | (1.77)* | (6.29)*** |
| EFIO | -0.004 | -0.038 | 0.025 | -0.008 | -0.049 | 0.064 |
|  | (-2.11)** | (-30.48)*** | (2.67)** | (-2.92)** | (-11.86)*** | (5.67)*** |
| LNY | 0.036 | 0.106 | 0.036 | 0.039 | 0.097 | 0.219 |
|  | (20.11)*** | (53.14)*** | (2.98)*** | (12.14)*** | (24.92)*** | (5.68)*** |
| LNPOPG | 0.013 | 0.046 | -0.066 | 0.045 | 0.072 | -0.051 |
|  | (3.22)*** | (27.47)*** | (-4.29)*** | (12.09)*** | (12.59)*** | (-9.70)*** |
| LNCPI | 0.093 | 0.083 | 0.104 | 0.054 | 0.043 | 0.071 |
|  | (52.30)*** | (54.48)*** | (17.43)*** | (16.51)*** | (12.13)*** | (9.26)*** |
| Constant | 3.794 | 3.351 | 3.024 | 3.158 | 3.158 | -0.580 |
|  | (114.47)*** | (157.38)*** | (15.99)*** | (11.57)*** | (20.33)*** | (-2.06)** |
| Wald chi2 | 3427.96*** | 8443.51*** | 429.93*** | 578.88*** | 2520.53*** | 14731.11*** |
| Sargan (Prob.) |  |  |  | 22.856 (1.00) | 23.344(1.000) | 36.752(1.000) |
| Autocorrelation | No | No | No |  |  |  |
| Obs. | 1704 | 744 | 960 | 1580 | 720 | 936 |

Note

*** indicates 1% significance level

** indicates 5% significance level

* indicates 10% significance level. lnHA = Hunger alleviation, lnFDI = foreign direct investment, lnAID = foreign aid, lnY = gross domestic product per capita, lnCPI = consumer price index, lnPOPG = population growth, EFIO = economic freedom index. GLS = Generalised least square with correlated disturbances, System GMM DPD = System generalised method of moment dynamic panel data estimator. Z-statistics are in parentheses.

In contrast, a percentage point improvement in the consumer price index could increase food production by 0.093%, 0.083%, and 0.104% at a 1% significance level across all samples, respectively. Interestingly, population growth seems not to benefit the least-developed countries. Our findings suggest that increasing population growth in those regions aggravates the efforts to increase food production. In contrast, a percentage point increase in population growth could strongly affect food production inversely by 0.066% at a 1% significance level. In contrast, the results for the all sampled group and developing countries sample revealed a contrary outcome. A percentage point increase in population growth in those regions positively leads to increased food production.

Statistically, it is imperative to robustly infer the outcome of findings by employing a robust method to that effect. However, we employed the system GMM dynamic panel data estimator to robust check the generalized least square results with correlated disturbances estimator. Table 6 presents the robust results alongside the primary estimator. Evidence from our robust estimations is consistently connected to the outcome of the primary estimator. Specifically, we observed that food production in both developing and least-developed countries as all sampled countries corrected itself in an absolute term by 18.5%. However, in developing and least-developed countries' samples, we witnessed 10.3% and 52.1%, correspondingly. To explain the other relationships, we conclude that foreign capital and food production are positively related.

**Table 7. Pairwise Dumitrescu Hurlin panel causality tests.**

| Pairwise Dumitrescu Hurlin Panel Causality Tests | All sample | | Developing countries | | Least-developed countries | |
|---|---|---|---|---|---|---|
| Null Hypothesis: | Zbar-Stat. | Direction | Zbar-Stat. | Direction | Zbar-Stat. | Direction |
| LNFDI & LNHA | 3.002** | ↔ | 1.557 | | 2.629* | ↔ |
| LNHA & LNFDI | 12.242*** | ↔ | 10.949*** | → | 6.671*** | ↔ |
| LNAID & LNHA | 2.009** | ↔ | -0.635 | | 3.236*** | ↔ |
| LNHA & LNAID | 8.565*** | ↔ | 3.962*** | → | 7.923*** | ↔ |
| EFIO & LNHA | 6.443*** | ↔ | 8.406*** | ↔ | 1.184 | |
| LNHA & EFIO | 15.698*** | ↔ | 3.556*** | ↔ | 17.784*** | → |
| LNGDPPC & LNHA | 13.580*** | ↔ | 15.081*** | ↔ | 4.816*** | ↔ |
| LNHA & LNGDPPC | 9.263*** | ↔ | 6.684*** | ↔ | 6.457*** | ↔ |
| LNCPI & LNHA | 12.597*** | ↔ | 9.805*** | ↔ | 8.152*** | ↔ |
| LNHA & LNCPI | 7.972*** | ↔ | 2.219** | ↔ | 8.668*** | ↔ |
| LNPOPG & LNHA | 10.806*** | ↔ | 8.373*** | ↔ | 7.025*** | ↔ |
| LNHA & LNPOPG | 25.187*** | ↔ | 10.610*** | ↔ | 24.217*** | ↔ |

Note

*** indicates 1% significance level

** indicates 5% significance level

* indicates 10% significance level. lnHA = Hunger alleviation, lnFDI = foreign direct investment, lnAID = foreign aid, lnY = gross domestic product per capita, lnCPI = consumer price index, lnPOPG = population growth, EFIO = economic freedom index. → denote unidirectional, ↔ denote bidirectional

All the estimators revealed robustness and fit-to-goodness to infer the outcomes of both estimators.

## 4.6 Dumitrescu-Hurlin panel causality test

In the study's quest to ascertain the causal linkage between the selected variables, the Dumitrescu Hurlin panel causality test was performed. Table 7 presents the outcome, and it can be reported that food production and foreign direct investment have bidirectional linkage in all sampled countries and Least-developed countries. Interestingly, all the independent variables exhibited bidirectional linkage with food production. Foreign aid showed a two-way causal relationship with food production in least-developed countries but showed unidirectional food production in developing countries. In the least developed countries, food production showed unidirectional causal linkage to economic freedom, but in developing countries, food production and economic freedom cause each other; thus, bidirectional linkage. Apart from these, all other variables showed a bidirectional causal linkage with each other.

## 5. Discussion

The study's findings highlight essential revelations concerning foreign capital, economic freedom, and food production. From the benchmark analysis of all sampled countries, we observed that foreign direct investment proportionally and significantly relates to food production same as foreign aid. However, economic freedom plays an inverse and significant role in such association. Dhahri and Omri [30] contend that foreign direct investment inflows for agricultural production are positively related. Moreover, they opined that foreign direct investment inflows into agricultural production ensure food security, promoting poverty alleviation in the long run. In support of these findings, Kaya et al. [58] observed that through improvements in managerial expertise and technological advancements from innovative contracts thanks to foreign direct investment inflows, food production efficiency is ensured by

overcoming the difficulties associated with lack of investments. In another study, Dhahri and Omri [31] confirmed a positive relationship between foreign direct investment and food production. Some scholars opined that the inflows of aids to developing countries could significantly relate to sustainable food production [55, 56]. That notwithstanding, Barkart and Alsamara [71] understand that total aid and agricultural aid positively relate to food production and agricultural output. This concludes our finding that total aid proportionally and significantly increases food production in low and middle-income countries.

According to the benchmark analysis, economic freedom plays an inverse and significant role in the nexus between foreign capital and food production. Despite the enormous benefits that emanate from economic freedom, the agricultural sector in developing and least-developed countries are characterised by the presence of externalities due to imperfect market competitions [29–31]. As a result, the markets cannot deliver efficient outcomes to propel food production. On the other hand, far less political participation, warehousing, derivatives markets, foreign trade, property rights on agricultural land, taxation of food production and national trade are involved in the path forward [71]. Futures market allows private individuals to research the food production sector and make potential price predictions. Moreover, governments should reassess their trade and tax policy choices and their possible effects immediately and work in concerted efforts to create an enabling climate for food trade and investments [4, 11, 12].

Moreover, Dhahri and Omri [30, 31] observed that foreign direct investment proportionally leads to agricultural production growth in developing and least-developed countries. According to Mekasha and Tarp [96] and Arndt et al. [97], foreign aid tends to increase the growth rate in an economy through investments into sectors that support growth [98]. Therefore, foreign aid could positively increase food production in least-developed countries. Total aid positively contributes to food production and could be attributed to the positive spillover effects from other sectors like energy, education, health, etc. [99]. According to the World Bank [34], improving road networks to rural areas, providing healthcare and education translates into people's improved wellbeing, mobility of labor, linking farms to markets, and high-quality labor to use sophisticated machinery to yield higher productivity.

## 6. Conclusion and policy implication

This present study assessed the impact of foreign capital (foreign aid and foreign direct investment) on food production in 31 developing and 40 least-developed countries. The study aimed to assess the impact of aggregate aid and foreign direct investment on this backdrop considering economic freedom's intervening role.

The study's findings suggest that foreign capital plays a positive role in sustainable food production. More importantly, economic freedom positively intervenes that relationship in least-developed countries in the long-run. On the other hand, the case is different in developing countries; economic freedom plays a negative and significant role in food production. The economic implication for this finding is that population growth rates have to be reduced. Consumer price index, economic freedom, and economic growth should be improved to support foreign capital (foreign aid and foreign direct investment) to contribute to food production substantially.

On the other hand, in developing countries, foreign capital plays a positive role in the long-run. However, economic freedom plays a negative role in sustainable food production in the long-run. The economic implication suggests that foreign capital should be increased proportionally with population, sustain economic freedom, and improve consumer price index towards growth to achieve sustainable food production. Moreover, developing countries

should not heavily rely on foreign aid but should seek foreign direct investment and optimize domestic resources to fund their sustainable food production expenditure. Notwithstanding that, policymakers and governments should create the enabling environment that would curtail growth-induced programs to ensure the needed sustainability in the food production sector.

The study acknowledges some limitation that could arise from the data used for economic freedom because most countries had no data in respect of some dimensions for measuring the overall economic freedom index. However, we recommend that future study should focus on the individual dimensions impact on food production. Also, the sectorial foreign investment inflows should be employed to ascertain the true impact of foreign investment channelled into food production.

## Supporting information

**S1 Appendix. List of Countries sampled for the study (Authors' own estimations).**
(DOCX)

## Author Contributions

**Conceptualization:** Guoping Ding, Prince Asare Vitenu-Sackey, Wenshu Chen, Xiaofeng Shi.

**Formal analysis:** Guoping Ding, Prince Asare Vitenu-Sackey, Xiaofeng Shi.

**Methodology:** Prince Asare Vitenu-Sackey, Jun Yan.

**Writing – original draft:** Prince Asare Vitenu-Sackey, Shichen Yuan.

**Writing – review & editing:** Guoping Ding, Prince Asare Vitenu-Sackey, Wenshu Chen.

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
