## [Decision Letter · Decision Letter 0]

22 Jun 2021

PONE-D-21-10241

The role of foreign capital and economic freedom in sustainable food production: Evidence from DLD countries

PLOS ONE

Dear Dr. Prince Asare Vitenu-Sackey,

Thank you for submitting your manuscript to PLOS ONE. After careful consideration, we feel that it has merit but does not fully meet PLOS ONE’s publication criteria as it currently stands. Therefore, we invite you to submit a revised version of the manuscript that addresses the points raised during the review process.

We look forward to receiving your revised manuscript.

Kind regards,

László Vasa, PhD

Academic Editor

PLOS ONE

Journal Requirements:

2. We note you have included a table to which you do not refer in the text of your manuscript. Please ensure that you refer to Table 4 in your text; if accepted, production will need this reference to link the reader to the Table.

Reviewers' comments:

Reviewer's Responses to Questions

**Comments to the Author**

1. Is the manuscript technically sound, and do the data support the conclusions?

Reviewer #1: Yes

Reviewer #2: Partly

2. Has the statistical analysis been performed appropriately and rigorously? 

Reviewer #1: Yes

Reviewer #2: I Don't Know

3. Have the authors made all data underlying the findings in their manuscript fully available?

Reviewer #1: Yes

Reviewer #2: Yes

4. Is the manuscript presented in an intelligible fashion and written in standard English?

Reviewer #1: Yes

Reviewer #2: Yes

5. Review Comments to the Author

Reviewer #1: The paper invetsigates an important field: how foreign investments affect the food production in certain countries. It is an essential topic, especially when emerging and developed countries protect their land and introduce a ban on purchasing arable land by foreigners.

The title, abstract, and keywords are correct. The title covers and reflects the content well.

The introduction is appropriate and well structured. The literature review is well detailed, critical, and analytical enough. However, here I found the only discrepancy in the paper actually: in the third part of the literature review, writing about economic freedom, here I feel an essential issue is missing: when talking about economic freedom, we should approach the different economic freedom indexes, at least the famous ones. So I recommend referring it and some works that are using this index for investigating the same field.

The methodology is well selected and described; it supports the results. Results are clearly demonstrated and allowed to conclude.

Reviewer #2: In my opinion, this is a well-constructed and extremely useful study, and the authors are excellent experts in the field they have just studied. What I see as a problem is that I only see the range of variables included in the study and not the data. Thus the research does not meet the criterion of being replicable by anyone. So my main comment is that the data table should be filled in. The other problem is that the text of the study has an extremely large number of complex tables, which makes it difficult to review the material. I suggest that what is not directly discussed should be included in the appendix.

6. PLOS authors have the option to publish the peer review history of their article (what does this mean?). If published, this will include your full peer review and any attached files.

Reviewer #1: No

Reviewer #2: No

---

## [Author Response · Author response to Decision Letter 0]

25 Jun 2021

Editor

1. Please ensure that your manuscript meets PLOS ONE’s style requirements, including those file naming. DONE

2. We note you have included a table to which you do not refer in the text of your manuscript. ALL TABLES AND FIGURE HAVE BEEN CITED IN THE TEXT OF THE MANUSCRIPT.

Reviewer 1

1. However, I here found the only discrepancy in the paper actually: in the third part of the literature review, writing about economic freedom, we should approach the different economic freedom indexes, at least the famous ones. So I recommend referring it and some works that are using this index for investigating the same field? OTHER STUDIES THAT USED THE ECONOMIC FREEDOM INDEXES HAVE BEEN DISCUSSED IN THE LITERATURE REVIEW PART. MOREOVER, TO THE BEST OF OUR KNOWLEDGE, NO STUDY HAS BEEN CONDUCTED CONSIDERING ECONOMIC FREEDOM WITH FOREIGN CAPITAL AND FOOD PRODUCTION – AND IN PARTICULAR USING THE ECONOMIC FREEDOM INDEX.

Reviewer Two

1. What I see as problem is that I only see the range of variables included and not the data. Thus the research does not meet the criterion of being replicable by anyone. THE DATA FOR THE STUDY HAS DEPOSITED AT MENDELEYDATA WITH A DOI-http://dx.doi.org/10.17632/fr2fcvftpj.1.

---

## [Decision Letter · Decision Letter 1]

12 Jul 2021

The role of foreign capital and economic freedom in sustainable food production: Evidence from DLD countries

PONE-D-21-10241R1

Dear Dr. Prince Asare Vitenu-Sackey ,

We’re pleased to inform you that your manuscript has been judged scientifically suitable for publication and will be formally accepted for publication once it meets all outstanding technical requirements.

Kind regards,

László Vasa, PhD

Academic Editor

PLOS ONE

Additional Editor Comments (optional):

Reviewers' comments:

Reviewer's Responses to Questions

**Comments to the Author**

1. If the authors have adequately addressed your comments raised in a previous round of review and you feel that this manuscript is now acceptable for publication, you may indicate that here to bypass the “Comments to the Author” section, enter your conflict of interest statement in the “Confidential to Editor” section, and submit your "Accept" recommendation.

Reviewer #1: All comments have been addressed

Reviewer #2: All comments have been addressed

2. Is the manuscript technically sound, and do the data support the conclusions?

Reviewer #1: Yes

Reviewer #2: Yes

3. Has the statistical analysis been performed appropriately and rigorously? 

Reviewer #1: Yes

Reviewer #2: Yes

4. Have the authors made all data underlying the findings in their manuscript fully available?

Reviewer #1: Yes

Reviewer #2: Yes

5. Is the manuscript presented in an intelligible fashion and written in standard English?

Reviewer #1: Yes

Reviewer #2: Yes

6. Review Comments to the Author

Reviewer #1: The authors improved the paper along the recommendations so I can accept it for publication in its curremt form.

Reviewer #2: The article has also improved compared to the previous version.In my opinion, this is a valuable article, congratulations to the authors.

7. PLOS authors have the option to publish the peer review history of their article (what does this mean?). If published, this will include your full peer review and any attached files.

Reviewer #1: No

Reviewer #2: No

---

## [Editor Report · Acceptance letter]

16 Jul 2021

PONE-D-21-10241R1 

The role of foreign capital and economic freedom in sustainable food production: Evidence from DLD countries 

Dear Dr. Vitenu-Sackey:

I'm pleased to inform you that your manuscript has been deemed suitable for publication in PLOS ONE. Congratulations! Your manuscript is now with our production department. 

Kind regards, 

on behalf of

Prof. Dr. László Vasa 

Academic Editor

PLOS ONE